# Healthcare Service Quality from the Point of Healthcare Providers' Perception at the Time of COVID-19

Olivera Ivanov [1,2,*], Zoran Gojković [1,3], Nenad Simeunović [4], Danijela Gračanin [4], Aleksandra Milovančev [1,5], Dejan Ivanov [1,6], Marko Bojović [1,2], Miloš Bugarčić [1] and Nikola Stojić [4]

1 Faculty of Medicine, University of Novi Sad, 21000 Novi Sad, Serbia; zoran.gojkovic@mf.uns.ac.rs (Z.G.); aleksandra.milovancev@mf.uns.ac.rs (A.M.); dejan.ivanov@mf.uns.ac.rs (D.I.); marko.bojovic@mf.uns.ac.rs (M.B.); milos.bugarcic92@gmail.com (M.B.)
2 Department for Radiation Oncology, Oncology Institute of Vojvodina, 21204 Sremska Kamenica, Serbia
3 Clinical Centre of Vojvodina, Department for Orthopedic Surgery and Traumatology, 21000 Novi Sad, Serbia
4 Faculty of Technical Science, University of Novi Sad, 21000 Novi Sad, Serbia; nsimeun@uns.ac.rs (N.S.); gracanin@uns.ac.rs (D.G.); nikola.stojic@arthaus.info (N.S.)
5 Department for Cardiology, Institute of Cardiovascular Diseases of Vojvodina, 21204 Sremska Kamenica, Serbia
6 Clinical Centre of Vojvodina, Department for Abdominal and Endocrine Surgery, 21000 Novi Sad, Serbia
* Correspondence: olivera.ivanov@mf.uns.ac.rs

**Abstract:** The pandemic of the Coronavirus 19 disease (COVID-19) has had significant impact on healthcare systems worldwide. The present study aims to investigate the service providers' quality dimensions in public sector hospitals in the Republic of Serbia during the COVID-19 pandemic and to propose a sustainable model for healthcare improvement. The study was conducted from September 2021 to December 2021. A modified SERPERF quality measurement questionnaire was distributed to healthcare workers in nine secondary care public hospitals of the Serbian Autonomous Province of Vojvodina (APV). Six hundred one questionnaires were found to be complete in all aspects and compared to 528 questionnaires from the database of the Provincial Secretariat for Health Care obtained from healthcare workers before the COVID-19 outbreak. The present study suggests that supportive measures during the COVID-19 pandemic are effective and, from the providers' perception, increase healthcare quality. Continual investment in healthcare would provide sustainable development of healthcare quality in the future, regardless of the pandemic conditions.

**Keywords:** Coronavirus disease 2019 (COVID-19); healthcare; sustainable improvement; service quality

## 1. Introduction

Two years after the Coronavirus-19 disease (COVID-19) pandemic outbreak, healthcare systems worldwide are still working hard to manage COVID-19 patients due to a lack of medical staff, medical equipment, few intensive care units' beds, lack of training and constantly changing and updating patient management protocols [1,2]. Human resources are a significant factor that contributes to the overall effectiveness of a healthcare system and affects its sustainability [3]. Most governments (e.g., England, France, Italy, Luxembourg, Romania, Bulgaria, Sweden, Finland, the Netherlands, Israel and some Cantons in Switzerland) have determined incentives to improve healthcare services, such as salary increases, new equipment, employment opportunities, training programs, etc. Service providers (doctors and nurses) experienced benefits such as salary increases, working with new equipment, the opportunity for employment and education, etc. during the COVID-19 outbreak, despite the fact that this period was characterized by more working hours and difficulties in patient management. The Republic of Serbia was well ranked by the Global Health Security Index (59/195 and 33/43 for the European region), which was supported by the quick applying of healthcare measures during the first wave of the

COVID-19 pandemic [3,4]. The Government of the Republic of Serbia conducted incentive measures in the past two years with the goal to stimulate healthcare workers in the period of crisis. The most significant measures were a salary increase, a COVID-19 bonus for overtime work, employment opportunities for doctors and nurses, new equipment, and three new full-equipped hospitals dedicated to COVID-19 patients' management that were built. Many awards and recognitions for healthcare workers were assigned by the President of the Republic of Serbia personally. It was assumed that the aforementioned measures would have a stimulative effect on healthcare workers and that service quality would possibly become better.

It was concluded by many studies that it is very important to establish a relevant and sustainable motivation system for healthcare providers to optimize their work quality regardless of the pandemic conditions [3,5,6]. Material and non-material rewards are equally significant. The Republic of Serbia has one of the lowest incomes of doctors and nurses in Europe. Therefore, a salary increase is of great importance for their motivation, but for the longer period, higher responsibility, self-actualization, importance and challenge in the management of COVID-19 patients is probably going to play a significant role.

Searching the literature, many models for healthcare service quality measuring were identified: Donabedian, SERVQUAL, HEALTHQUAL, PubHosQualiHospitalQual, etc. [7–12]. The most widely used scale is SERVQUAL [13,14], which consists of five essential service quality dimensions:

1. Tangibles
2. Reliability
3. Responsiveness
4. Assurance
5. Empathy

Cronin and Taylor have developed the SERVPERF (Service Performance) scale, which represents an improved version of the SERQUAL model, and which was created on the basis of a critique of SERQUAL [15]. They claimed that the SERVPERF measures have greater validity because of "their content and discriminant validity, and they present a lengthy comparison of the convergent and discriminant characteristics". Actually, SERPERF presents a performance based approach to the measurement of service quality, which is more appropriate for healthcare service.

The aim of this study was to analyze whether government incentives have an impact on the quality of healthcare from the point of view healthcare providers' perception (physicians and nurses) compared to their quality perception in the working period before the pandemic. Second, we aimed to identify the most vulnerable quality dimension in a healthcare setting in order to propose future directions in the improvement of healthcare quality.

## 2. Materials and Methods

A cross-sectional multicentric study was conducted from September 2021 to December 2021. The modified SERPERF quality measurement anonymous questionnaire (Appendix A) was distributed to healthcare workers in nine secondary care public hospitals of APV, on a voluntary basis, during the COVID-19 outbreak from September 2021 to December 2021. The Institutional Review Board Statement was obtained from each hospital involved in the study. The study was carried out in accordance with ethical standards of the 1964 Helsinki Declaration, and participants were informed about all relevant aspects of the study. Patient consent was waived due to the fact that questionnaires used in the study were completely anonymous. Respondents did not need to provide their name, signature or any other personal identification data. Data were collected from 717 respondents working in nine secondary care public hospitals of APV (medical doctors and nurses), out of which 601 questionnaires were found complete in all aspects (COVID-19 group). One hundred sixteen questionnaires were incomplete; respondents did not answer all questions. In the next step, collected data were compared to 528 questionnaires from the database of the Provincial Secretariat for Health Care obtained previously from healthcare workers

of nine hospitals of APV before the COVID-19 outbreak (non-COVID-19 group). These questionnaires were part of the survey conducted from September 2019 to December 2019, as a part of the quality measurement initiated by the Provincial Secretary for Healthcare, and the same modified SERPERF quality measurement anonymous questionnaire was used as well. One of the activities of the Provincial Secretary for Healthcare is measuring quality in the healthcare system in non-pandemic circumstances. The questionnaire used for the purpose of this study was identical to the one used previously for the mentioned purpose and the same methodology was used. There was curiosity of if healthcare workers' answers would differ significantly during the pandemic time.

Demographic characteristics of respondents, such as age, sex, education level and amount of income, were collected. The SERVPERF methodology was used for measuring the quality of service from the point of view of healthcare workers with five service quality dimensions. For the study purpose, the original SERPERF questionnaire was extended from 22 to 26 questions to emphasize healthcare workers' motivation. Within each dimension, there were several items (26 in total) measured on a five-point scale from strongly agree to strongly disagree (Appendix A). Two groups were compared by each question and dimension separately: tangibility, reliability, responsiveness, assurance and empathy. Finally, two groups were compared by all dimensions together.

*Data Analysis*

Descriptive statistics are presented as percentages and the minimum and maximal values. A Chi-squared test was used to compare age, sex, amount of income and education level between two groups. A Man–Whitney U test was used to compare median values for each question among all five dimensions between two groups. Mean values were used to compare dimensions between two groups. A *p*-value less than 0.001 was considered statistically significant. The statistical analysis was performed using SPSS 23.0 and Jamovi V2.2.2 statistical software.

## 3. Results

The study included 1129 respondents in total, aged 22–65 years. Out of them, 601 respondents were in the COVID-19 group and compared to 528 respondents in the non-COVID-19 group. Regarding age, respondents were divided in five groups (less than 30-years, in range from 31–40 years, 41–50 years, 51–60 years and more than 60 years) and compared between two groups. There was a statistically significant difference in two age ranges when groups were compared; in the COVID-19 group there were more respondents in the 41–50 years range (34.8% vs. 26.1%, $p < 0.001$). On the contrary, less respondents were in the COVID-19 group in group of 51–60 years (19.6% vs. 25.5%, $p < 0.001$). In other age groups, there was no statistical difference observed. There was no statistical significance in age distribution when whole groups were compared ($p = 0.468$ Table 1).

**Table 1.** Age distribution and amount of income in two observed groups.

|  |  | Non-COVID-19 Group | COVID-19 Group |
|---|---|---|---|
| **Age (year)** | min | 22 | 20 |
|  | max | 63 | 65 |
|  | mean | 43.6 | 44.1 |
|  | SD | 11.6 | 11.1 |
|  | *p* | 0.468 [a] | |
| **amount of income (€)** | min | 230 | 190 |
|  | max | 1410 | 1650 |
|  | mean | 568.20 | 866.22 |
|  | SD | 265.0 | 371.9 |
|  | *p* | >0.001 [a] | |

[a]—Independent Samples T test; SD—Standard Deviation.

There was a significant difference between the two groups in sex, as more female respondents were identified in the COVID-19 group compared to non-COVID-19 group (70.3% vs 87.9%, *p* < 0.001).

According to the amount of income, respondents in both groups were divided in nine groups: monthly less than 200 €, from 200–300 €, from 300–400 €, from 400–500 €, from 500–600 €, from 600–700 €, from 700–800 €, from 800–1000 € and more than 1000 €. There was a statistically significant difference between the COVID-19 and non-COVID-19 group in the six income groups according to the amount of income (Chi square test, *p* < 0.001). Less respondents were observed in the COVID-19 group in the following mainly lower-income groups: 300–400 € (0.3% vs. 36.9%), 400–500 € (2.8% vs. 18.6%), 700–800 € (3.3% vs. 11.7%) and 800–1000 € (12.1% vs. 16.7%). A salary in amount from 500–600 € and more than 1000 € was significantly more prevalent in respondents in COVID-19 compared to respondents before the pandemic (27.6% vs. 2.3% and 32.6% vs. 6.8%). There was no statistical significance in the amount of income when the whole COVID-19 and non-COVID-19 groups were compared (*p* > 0.001 Table 1). There was no statistical significance in education when groups were compared (*p* = 0.122).

Among each dimension, there were several questions: responsiveness (5), assurance (5), reliability (4), empathy (7) and tangibility (5). Answers to each question were compared between the COVID-19 and non-COVID-19 group. The Man–Whitney U test showed a statistically significant difference when all answers among five dimensions of service providers' quality were compared between the COVID-19 and non-COVID-19 group (*p* < 0.001), except in one question (dimension tangibility, see Appendix A, Table A2, D2.1) (*p* = 0.417). However, answers to the other four questions for dimension tangibility were significantly different between the two groups. The respondents in both groups accorded the highest priority to responsiveness, followed by assurance, reliability, empathy and then tangibility. Responsiveness was significantly more assessed in the COVID-19 group: 4.38 vs. 4.12 in the non-COVID-19 group (Figure 1a). There was also a significant difference between the two groups in assurance dimension, with a median mark of 4.34 in the COVID-19 vs. 4.07 in the non-COVID-19 group (Figure 1b). There was a significant difference observed between two groups in the reliability dimension, with a mean value of 4.33 in the COVID-19 vs. 4.13 in the non-COVID-19 group (Figure 1c). In the COVID-19 group, respondents reported more empathy in the COVID-19 group (4.05) compared to non-COVID-19 respondents (3.82) (Figure 1d). Finally, in the COVID-19 group, healthcare workers felt more tangible (3.90) compared to in the non-COVID group (3.64) (Figure 1e). Tangibility was identified as the most vulnerable quality dimension of a healthcare setting in this study. The radial diagram in Figure 2 presents a comparison of both groups regarding all five dimensions.

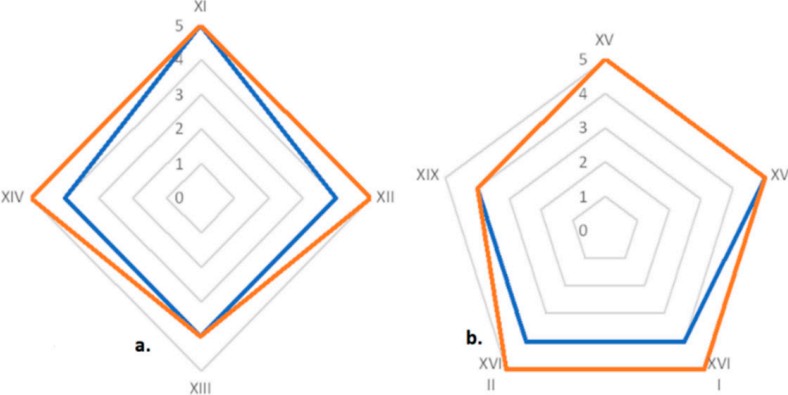

**Figure 1.** *Cont.*

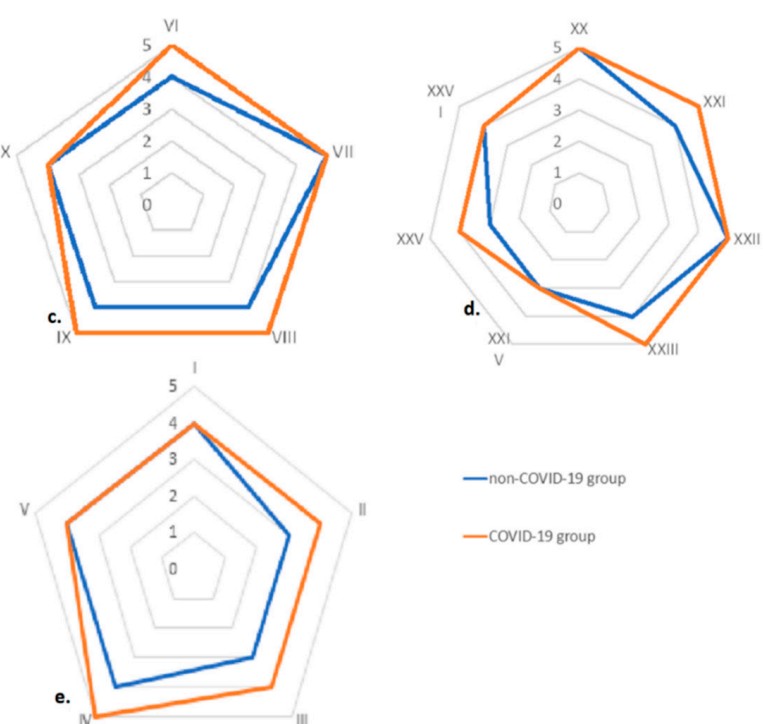

**Figure 1.** Comparison of the dimensions of responsiveness (**a**), assurance (**b**), reliability (**c**), empathy (**d**) and tangibility (**e**) between the COVID-19 and non-COVID-19 group.

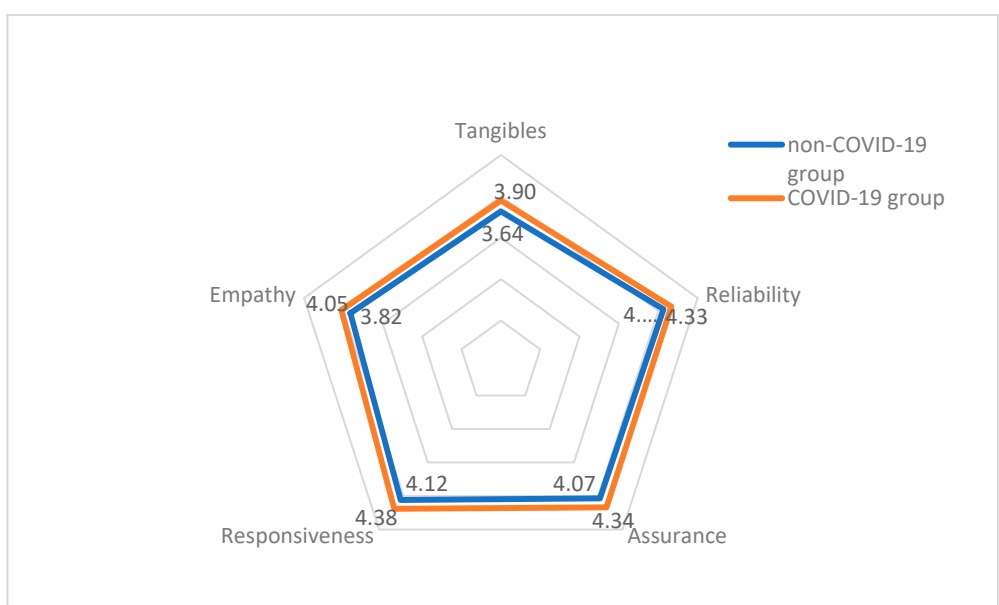

**Figure 2.** Comparison of all five dimensions between the COVID-19 and non-COVID-19 group.

## 4. Discussion

The COVID-19 outbreak has significantly influenced all aspects of life. Providing medical care was challenging for healthcare workers, who were faced with high-pressure work, stress, long working hours and a lack of equipment and intensive care unit's beds.

One of the main goals of sustainable development of a healthcare system is to achieve Universal Health Coverage (UHC), including protection from financial risks and availability of basic healthcare services [16]. However, a high quality of healthcare is challenging in many environments, and there is an urgent need for activities that will lead to general improvement in this field, especially in pandemic circumstances. The World Health Organi-

zation, World Bank and OECT have proposed several actions for governments, healthcare service providers and healthcare customers who should work together to reach the goal of a high quality health care system. When the pandemic crisis began, Governments from different countries proposed incentive measures to help healthcare workers to cope with many difficulties in their work (e.g., England, France, Italy, Luxembourg, Romania, Bulgaria and Sweden) [17]. For example, Finland, the Netherlands and some Cantons in Switzerland provided special budgets for hospitals, and Germany provided higher per diem (PD) rates for hospitals with intensive care units. Hospitals in Israel received an additional budget as well to compensate for the loss from the decrease in medical tourism. The Government of Republic of Serbia has provided an additional budget for COVID-19 healthcare workers and new equipment, as mentioned before. However, the interesting point was how to measure the effect of governments' measures on healthcare workers and service quality?

Although SERQUAL was widely used by a vast array of service industries, it was criticized by many authors [18–22]. Inappropriateness for measuring healthcare service quality was addressed. Cronin and Taylor then came up with SERVPERF, as they believed that service quality is an antecedent to customer satisfaction [23]. A further study conducted by Paul (2003) came up with conclusion that SERPERF is superior and more convenient compared to SERVQAL [24]. Other authors have also proposed different adaptations of the SERQUAL model, with limited data about its applicability [25]. In general, it was concluded that each country should develop a personalized quality measurement scale that would adapt to special social environments [25–27]. For the study purpose, the original SERPERF questionnaire was extended from 22 to 26 questions to emphasize healthcare worker's motivation, which is the most important issue in the pandemic setting. It was assumed that motivation is mainly influenced by incentive measures conducted by the Government. It has been concluded by many studies that it is very important to establish a relevant and sustainable motivation system for healthcare providers to optimize their work quality regardless of the pandemic conditions [3,28]. To the best of our knowledge, this a unique study in the literature, which analyzed healthcare quality from the aspect of healthcare workers via five quality dimensions and the SERPERF methodology. Salary increase is one of the most important motivating factors, according to results of a recently conducted meta-analysis [3,29]. The Republic of Serbia has one of the lowest incomes of physicians and nurses in Europe. Therefore, a salary increase was of great importance for their motivation, and results of our study show that there was a statistically significant difference between two groups. A salary in the amount from 500–600 € and more than 1000 € was significantly more prevalent in respondents at COVID-19 compared to respondents before the pandemic (27.6% vs. 2.3% and 32.6% vs. 6.8%). Pay has meaning beyond being able to buy goods: it has the psychological effect of reward, and our results confirm this fact, as it was analyzed in the literature [30,31]. However, some authors describe that rewards may lead to lower performance than no rewards in certain circumstances, which implies that motivation is a complexity of cultural values, personal environment and individual psychological issues. In light of these conclusions, for the longer period, higher responsibility, autonomy, self-actualization, scope to use and develop skills and abilities, importance and challenge in the management of COVID-19 patients could possibly play significant roles for the sustainable quality of healthcare services.

Analyzing five quality measurement dimensions, the respondents in both groups accorded the highest priority to responsiveness. Respondents in both groups answered that, in their opinion, doctors and nurses are prompt in their response, polite and willing to help. They gave the highest marks to this dimension in favor of the COVID-19 group (4.38 vs. 4.12). Comparing answers to all questions among the dimensions of responsiveness, a statistically significant difference was observed in favor of the COVID-19 group ($p < 0.001$). It is obvious that the pandemic situation caused high agility in the service of medical staff who considered patients' management as the highest priority, which should be done promptly without any delay. It is possible that the highest priority of the dimension of responsiveness observed in this study was a consequence of the pandemic, when prompt

medical intervention is very important for treatment outcome. In the searched literature, there are not any data analyzing the dimension of responsiveness from the point of health care workers as service providers. However, there are data that emphasize the role of responsiveness from the point of view of service users (patients) [25].

Healthcare providers consider the assurance of healthcare services as being the second most important factor for service quality. The assurance dimension combines several factors, such as competence, courtesy, credibility and security, and it refers to the politeness and respect and that medical staff have requisite skills and knowledge. Analyzing this dimension, healthcare workers in both groups have a perceived high level of their knowledge and competence (median marks 4.34 vs. 4.07). However, a statistically significant difference was observed in favor of the COVID-19 group when answers to all questions were compared separately ($p < 0.001$). They described their relationship with patients as "full of trust", and in their opinion, patients feel safe and well informed. Ranking of assurance among service quality dimensions varied among different authors, from first to fourth place [32].

The dimension of reliability was ranked in the third place in both groups (4.33 vs. 4.13), with a statistically significant difference in marks to each question in favor of theCOVID-19 group ($p < 0.001$). In the available literature, there are no data analyzing this dimension from the point of view of healthcare providers, but from the aspect of patients, this dimension appears to be the most significant [33]. Reliability has been classed as the first dimension of the SERVQUAL service quality model [32].

The respondents in both groups feel that they understand the needs of the patients, with personalized attention and understanding of their problems. In their opinion, patients feel supported during the hospital stay. Although the dimension of empathy was better marked in the COVID-19 group (4.05 vs. 3.82) with a statistical significance when answers to each question were compared separately ($p < 0.001$), it is clear that in general, empathy should be at a higher level. However, the fast-track of patients did not make this dimension worse compared to non-pandemic conditions. In the searched literature, the dimension of empathy was ranked poorly as well [33].

In this study, tangibility was answered with the lowest mark in both groups (3.90 vs. 3.64), but the statistical difference was in favor of the COVID-19 group in all answers except one. In other words, the pandemic conditions did not influence this dimension; moreover, healthcare workers felt tangibility on a higher level compared to non-COVID-19 circumstances. In the literature, tangibility was also ranked as a dimension on a low hierarchy level [32]. Tangibility was identified as the most vulnerable quality dimension of a healthcare setting in this study, which implicates the need for upgrading the appearance and behavior of the medical staff via different workshops and hospital measurements. As a result, a higher degree of tangibility involved in the provision of care should be obtained.

In the current study, there are few limitations. First, the study after the outbreak was confined to four months, which is a short period to conclude if the improvement of healthcare quality is sustainable. Further investigations, including a longer study period, are needed to answer this question. Second, healthcare workers' emotional state and mental health were not taken into consideration.

## 5. Conclusions

The outbreak of COVID-19 in most European countries forced healthcare systems to reassess its quality parameters. Furthermore, the pandemic represents a unique circumstance that will need to be investigated with more extensive studies with refined methodology. Incentive measures are very effective, influence healthcare workers' motivation and increase healthcare service quality as perceived by physicians and nurses. The present study suggests that supportive measures during the COVID-19 pandemic are effective and, from the providers' perception, increase healthcare quality. The rapid and prompt adjustment of payment and other stimulation of healthcare workers, including new equipment, is essential for a sustainable healthcare system in challenging times.

**Author Contributions:** Conceptualization, N.S. (Nenad Simeunović) and O.I.; methodology, D.G.; software, A.M.; validation, Z.G., O.I. and N.S. (Nenad Simeunović); formal analysis, D.I.; investigation, M.B. (Marko Bojović); resources, M.B. (Marko Bojović); data curation, M.B. (Miloš Bugarčić); writing—original draft preparation, O.I.; writing—review and editing, O.I.; visualization, N.S. (Nikola Stojić); supervision, D.I.; project administration, M.B. (Miloš Bugarčić); funding acquisition, O.I. All authors have read and agreed to the published version of the manuscript.

**Funding:** This research received no external funding.

**Institutional Review Board Statement:** The Institutional Review Board Statement was obtained from each hospital involved in the study: OB "dr Radivoj Simonović Sombor protocol code 23-4919,16.9.2021; OB Sremska Mitrovica protocol code 6553/4,15.9.2021; OB Kikinda protocol code 01-264/1,15.9.2021; OB Senta protocol code 51-1388,17.9.2021; OB Pančevo protocol code 01-20/2021,15.9.2021; OB Subotica protocol code 01-6389,15.9.2021; OB Vrbas protocol code 316/3/2021, 14.9.2021; OB Vršac protocol code 01-1046,17.9.2021; OB "Đorđe Joanović" protocol code 01-1001/21,17.9.2021. Zrenjanin.

**Informed Consent Statement:** Patient consent was waived due to the fact that questionnaires used in the study were completely anonymous. Respondents did not need to provide their name, signature or any other personal identification data.

**Data Availability Statement:** The Institutional Review Board Statement was obtained from each hospital involved in the study.

**Conflicts of Interest:** The authors declare no conflict of interest.

## Appendix A

**Table A1.** Sociodemographic characteristics of respondents.

| Title 1 | Title 2 | Title 3 |
|---------|---------|---------|
| D1.1 | Gender *(circle)* | Male<br>Female |
| D1.2 | Age *(circle)* | Under 30 years of age<br>From 31 to 40 years of age<br>From 41 to 50 years of age<br>From 51 to 60 years of age<br>Over 60 years of age |
| D1.3 | Level of education *(circle)* | Elementary school<br>High school lasting 3 years<br>High school lasting 4 years<br>College (vocational studies)<br>University (Bachelor's degree)<br>University (Master's degree)<br>University (PhD) |
| D1.4 | City of residence *(write)* | |
| D1.5 | Job *(circle)* | Nurse<br>MD general practitioner<br>MD specialist |
| D1.6 | Monthly salary—net *(circle)* | Under 20,000 RSD<br>From 20,000 to 30,000 RSD<br>From 30,000 to 40,000 RSD<br>From 40,000 to 50,000 RSD<br>From 50,000 to 60,000 RSD<br>From 60,000 to 70,000 RSD<br>From 70,000 to 80,000 RSD<br>From 80,000 to 100,000 RSD<br>Over 100,000 RSD |
| D1.7 | Marital status *(circle)* | Not married<br>Married<br>Divorced<br>Widow/widower |
| D1.8 | Institution/hospital of workplace *(write)* | |
| D1.9 | Ward *(write)* | |

**Table A2.** Perception of the staff of the quality of given health services in the health institution.

| No | Description | Absolutely Disagree | | Partially Agree | | Absolutely Agree |
|---|---|---|---|---|---|---|
| D2.1 | 1. In the health institution I work in, the staff take care of the hygiene in the areas where patients are treated. | 1 | 2 | 3 | 4 | 5 |
| D2.2 | 2. The surroundings in the health institution I work in are comfortable enough for the patients to rest. | 1 | 2 | 3 | 4 | 5 |
| D2.3 | 3. The health institution I work in has good and modern equipment for treating patients. | 1 | 2 | 3 | 4 | 5 |
| D2.5 | 4. The health staff in the health institution I work in are tidy and professional. | 1 | 2 | 3 | 4 | 5 |
| D2.6 | 5. In the health institution I work in, patients receive adequate material and information about the health care they receive (i.e., brochures). | 1 | 2 | 3 | 4 | 5 |
| D2.7 | 6. The medical staff in the health institution I work in give the patients precise service with the knowledge of all needed skills. | 1 | 2 | 3 | 4 | 5 |
| D2.8 | 7. Patients receive all necessary information about treatment and are asked for certain permissions when they are needed. | 1 | 2 | 3 | 4 | 5 |
| D2.10 | 8. The medical staff in the health institution I work in are dependable. | 1 | 2 | 3 | 4 | 5 |
| D2.11 | 9. The health service is given to the patients within the time told to them. | 1 | 2 | 3 | 4 | 5 |
| D2.12 | 10. The health institution I work in provides excellent effects in treating its patients. | 1 | 2 | 3 | 4 | 5 |
| D2.14 | 11. When the patient needs help, the medical staff in the health institution I work in always willingly help. | 1 | 2 | 3 | 4 | 5 |
| D2.16 | 12. The patients receive the services in time, as well as necessary medications. | 1 | 2 | 3 | 4 | 5 |
| D2.17 | 13. The health institution I work in has a simple system for making an appointment. | 1 | 2 | 3 | 4 | 5 |
| D2.18 | 14. The medical staff in the health institution I work in willingly answer to the questions of the patients. | 1 | 2 | 3 | 4 | 5 |
| D2.19 | 15. The medical staff in the health institution I work in have the knowledge to give needed services. | 1 | 2 | 3 | 4 | 5 |
| D2.20 | 16. Patients are given all necessary information about the hospitalization (when it is needed) in the health institution I work in. | 1 | 2 | 3 | 4 | 5 |
| D2.22 | 17. The medical staff in the health institution I work in give the patients services with a sense of duty. | 1 | 2 | 3 | 4 | 5 |
| D2.23 | 18. Patients feel safe when interacting with the medical staff in the health institution I work in. | 1 | 2 | 3 | 4 | 5 |
| D2.24 | 19. The buildings of the health institution I work in are safe. | 1 | 2 | 3 | 4 | 5 |
| D2.26 | 20. The medical staff in the health institution I work in respect the individual personalities of the patients. | 1 | 2 | 3 | 4 | 5 |
| D2.27 | 21. The patients' complaints are heard and listened to in the health institution I work in. | 1 | 2 | 3 | 4 | 5 |
| D2.29 | 22. The privacy of the patients is respected in the health institution I work in. | 1 | 2 | 3 | 4 | 5 |
| D2.30 | 23. Care is taken with the families of the patients and those coming to visit them in the health institution I work in (in case the patient is held in for therapy). | 1 | 2 | 3 | 4 | 5 |
| D2.31 | 24. In the health institution I work in, the biggest system of motivation to give the patients better services is the salary. | 1 | 2 | 3 | 4 | 5 |
| D2.32 | 25. In the health institution I work in, the biggest system of motivation to give the patients better services is the modern equipment. | 1 | 2 | 3 | 4 | 5 |
| D2.33 | 26. In the health institution I work in, the biggest system of motivation to give the patients better services is the possibility of further education of the staff through seminars and symposiums. | 1 | 2 | 3 | 4 | 5 |

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
