# Peer review of "Healthcare Service Quality from the Point of Healthcare Providers’ Perception at the Time of COVID-19"

_challenges, doi:10.3390/challe13010026_

Round 1

Reviewer 1 Report

Comments

Introduction
-    Were the healthcare workers’ emotional state and mental health taken into consideration in the measures used?

Materials and Methods
-    Typo of period after time periods for example ‘September 2021.’. Please amend for example ‘September 2021.’ to ‘September 2021’ in lines 80 and 96
-    Shouldn’t consent still be obtained even though anonymity was maintained? I believe this would and should still be attainable
-    How was missing data managed?
-    Why was age not analysed as a continuous variable and means compared? Was there a rationale for categorising age groups? I don’t think Figure 1 and 2 are strictly necessary as it doesn’t add any additional information to the written data. Both of these can be presented in table form instead. 

Discussion
-    Suggested grammatical corrections: line 167 ‘Providing medical care was challenged for healthcare workers…’ to ‘Providing medical care was challenging for healthcare workers…’. Suggest to change line 173 ‘there is permanent need for activities’ to ‘there is an urgent need for activities…’. Suggest to change line 177 ‘When pandemic crisis has begun…’ to ‘When pandemic crisis began…’

Author Response

Introduction
-    Were the healthcare workers’ emotional state and mental health taken into consideration in the measures used?

According to the method used in this study, only opinion of healthcare workers about different service aspects has been taken into account and analyzed. Therefore, personal data about respondents, including their emotional state and mental health, were not collected for the purpose of this study. However, emotional state and mental health of healthcare workers are also very important issues and could be analyzed in future studies, thank You for Your comment.

Also included in limitation.

Materials and Methods
-    Typo of period after time periods for example ‘September 2021.’. Please amend for example ‘September 2021.’ to ‘September 2021’ in lines 80 and 96

Correction was made, now line 60 and 76

-    Shouldn’t consent still be obtained even though anonymity was maintained? I believe this would and should still be attainable

For the study purpose, anonymous questionnaire was distributed to the healthcare workers of 9 hospitals. All medical staff were informed verbally about all relevant aspects of the study. If they decided to participate, they collected the questionnaire from the information desk, and after answering the questions they returned it to the same place. Furthermore, in questionnaire form short written explanation of its purpose was provided.  Patient consent was waived due to the fact that respondents didn’t need to provide their name, signature or any other personal identification data. Thank You for Your comment.

-    How was missing data managed?

In the analyzed questionnaires, there were no missing data. All of the incomplete questionnaires were excluded from the analysis. Only complete forms were taken into account.

-    Why was age not analysed as a continuous variable and means compared? Was there a rationale for categorising age groups? I don’t think Figure 1 and 2 are strictly necessary as it doesn’t add any additional information to the written data. Both of these can be presented in table form instead. 

There was a rationale for categorizing age groups, due to the fact that healthcare workers have different degree of responsibility, job position, salary etc. according to their age / work experience. We aimed to exclude influence of these factors on study results and to avoid bias if, for example, in one of the groups dominate one or two age groups. However, we added continuous variables in the Table 1. And compared between groups. The same analysis was made for amount of income. We excluded Figure 1. And 2.

Discussion
-    Suggested grammatical corrections: line 167 ‘Providing medical care was challenged for healthcare workers…’ to ‘Providing medical care was challenging for healthcare workers…’.

Correction was made, now line 163

Suggest to change line 173 ‘there is permanent need for activities’ to ‘there is an urgent need for activities…’. Correction was made, now line 168

Suggest to change line 177 ‘When pandemic crisis has begun…’ to ‘When pandemic crisis began…’

Correction was made, now line 172

Reviewer 2 Report

The pandemic has given rise to many opportunities for research. Apart from clinical trials, there are other types of research which produce very useful data for new services, quality improvement and health service management. The researchers of the study reported in the manuscript took the opportunity to examine the quality dimensions and supportive measures from the perspective of the providers. Although the study was conducted in the Republic of Serbia, I am confident that the findings are applicable to many countries in the world where nurses and doctors are relatively underpaid.

There are several areas that require further work from the researchers.

(1) Research method - More description on the procedures would help readers understand the study particularly the similarity and difference between the data collection procedure before COVID-19 outbreak and after the outbreak. Was the context the same? 

(2) Consent form - While I agreed consent form could be waived as the action of filling in the questionnaire implied consent, I believe the reason behind it should not be anonymity. In most clinical studies, the subjects are kept anonymous but they still need to consent after the researcher/s explain clearly to them what they will go through. 

(3) Questionnaire - There are some errors in the section about the sociodemographic characteristics of the subjects. For example, under Title 2, the options for Workplace are jobs or positions. For the monthly salary, there are overlapping options. If a person earns 30.000 RSD, he/she can choose either 'From 20.000 to 30.000 RSD' or 'From 30.000 to 40.000 RSD'. This also happens to the other options. 

(4) Data analysis - The researchers chose p < 0.001 instead of the one we normally use, p ≤ 0.05. It would help if the researchers explain the rationale behind. There are some mistakes in the legend of the graph (Figure 1). Overlapping values are also noted, which happens in Figure 2 too.

(5) The radial diagrams in Figure 3 are too small to read.

(6) The study after the outbreak was confined to four months. This may be considered too short for any major improvement to occur particularly if we want to know whether the improvement is sustainable. Hence, a section on limitations of the study is needed. 

(7) There is no explicit section for conclusion in this manuscript. I think it is necessary and essential for the manuscript to complete its mission. 

Author Response

  • Research method - More description on the procedures would help readers understand the study particularly the similarity and difference between the data collection procedure before COVID-19 outbreak and after the outbreak. Was the context the same? 

Line 78 added: One of the activities of Provincial Secretary for Healthcare is measuring quality in healthcare system in non-pandemic circumstances. The questionnaire used for the purpose of this study was identical to the one used previously for the mentioned purpose and the same methodology was used. There was curiosity if healthcare workers’ answers would differ significantly during the pandemic time.   

  • Consent form - While I agreed consent form could be waived as the action of filling in the questionnaire implied consent, I believe the reason behind it should not be anonymity. In most clinical studies, the subjects are kept anonymous but they still need to consent after the researcher/s explain clearly to them what they will go through. 

For the study purpose, anonymous questionnaire was distributed to the healthcare workers of 9 hospitals. All medical staff were informed verbally about all relevant aspects of the study. If they decided to participate, they collected the questionnaire from the information desk, and after answering the questions they returned it to the same place. Furthermore, in questionnaire form short written explanation of it purpose was provided.  Patient written consent was waived due to the fact that respondents didn’t need to provide their name, signature or any other personal identification data. But, I completely agree that they still need to consent after clear explanation. Thank You for Your comment.

  •  
  • Questionnaire - There are some errors in the section about the sociodemographic characteristics of the subjects. For example, under Title 2, the options for Workplace are jobs or positions. For the monthly salary, there are overlapping options. If a person earns 30.000 RSD, he/she can choose either 'From 20.000 to 30.000 RSD' or 'From 30.000 to 40.000 RSD'. This also happens to the other options. 

It was rare that one has exactly 30.000 or 40.000 but if they did, they didn’t answer or circle both options. We excluded from the study all these questionnaires. Perhaps, If You suggest, we can put ranges from 30.000-39.999, and 50.000-59.999, etc. Please comment on that, and thank You very much.

  • Data analysis - The researchers chose p < 0.001 instead of the one we normally use, p ≤ 0.05. It would help if the researchers explain the rationale behind. There are some mistakes in the legend of the graph (Figure 1). Overlapping values are also noted, which happens in Figure 2 too.

The traditional reporting of P values (indicating only that P<0.05) simply indicated whether the results were "statistically significant" or not. But P values of 0.051 and 0.049 should be interpreted similarly despite the fact that the 0.051 is greater than 0.05 and is therefore not "significant" and that the 0.049 is less than 0.05 and thus is "significant." Reporting actual P values avoids this problem of interpretation, and choosing p < 0.001 might provide higher level of preciseness.

As one of the reviewer recommended, we excluded figure 1. and figure 2. from the manuscript, instead of that tables are presented.

  • The radial diagrams in Figure 3 are too small to read.- enlarged

  • The study after the outbreak was confined to four months. This may be considered too short for any major improvement to occur particularly if we want to know whether the improvement is sustainable. Hence, a section on limitations of the study is needed. 

In the current study, there are few limitations. First, the study after the outbreak was confined to four months which is short period to conclude if the improvement of healthcare quality is sustainable. Further investigations, including longer study period are needed to answer this question. Second, healthcare workers’ emotional state and mental health were not taken into consideration.

  • There is no explicit section for conclusion in this manuscript. I think it is necessary and essential for the manuscript to complete its mission. 
  1. Conclusion

The outbreak of COVID-19 in most European countries forced healthcare systems to reassess its quality parameters. Furthermore, pandemic represents a unique circumstance that will need to be investigated with more extensive studies with refined methodology. Incentive measures are very effective, influence healthcare workers’ motivation and increase healthcare service quality as perceived by physicians and nurses. The present study suggests that supportive measures during COVID-19 pandemic are effective and from the providers’ perception increase healthcare quality. Rapid and prompt adjustment of payment and other stimulation of healthcare workers including new equipment is essential for sustainable healthcare system in challenging times.

Reviewer 3 Report

Abstract

1) Please define SERPERF. 

Introduction

2) Line 37 "Most governments have determined incentives", How many? please provide a number and some names as example.

3) Line 39 "Service providers (doctors and nurses) experienced some benefits", Which benefits? please provide examples.

4) Line 41 "The Republic of Serbia was well", please provide de number in the ranking. 

5) Line 47 "and 3 new full-equipped " change it por "and three new full-equipped". 

6) Line 52 "It is concluded by many studies ...". Concluded by whom? if they conclude that, what is the novelty of this paper?

Results

7) Provide the central value and dispersion value of the age of both groups, also compare it with a T-student o Mann-Whitney U test (depending on their distribution). Is more powerful to compare the age as a continuous variable than a categorical. Also change the Figure 1 for continuous data. 

8) Same modifications for the income that are asked in the point #7 (for text and figure), es more accurate the analysis of the continuous data than the income as a categorical data. 

9) Make a more clear, deep and critical interpretation of the results from the line 143-158.

10) Improve de quality of the figure 3, the legends are not clear. 

Discussion

11) Make a more comparative analysis also provide a deeper explanation of the consequences of the results. The discussion is very superficial.

Author Response

  1. 1) Please define SERPERF. 

Line 47 Cronin and Taylor have developed SERVPERF (Service Performance) scale, which represents improved version of SERQUAL model and was created on basis of critique of SERQUAL [15].

Introduction

2) Line 37 "Most governments have determined incentives", How many? please provide a number and some names as example.

The exact number cannot be provided, probably each country have determined some measures, it is hard to estimate , examples are given in the text(e.g. England, France, Italy, Luxembourg, Romania, Bulgaria, Sweden, Finland, the Netherlands, Israel and some Cantons in Switzerland)

3) Line 39 "Service providers (doctors and nurses) experienced some benefits", Which benefits? please provide examples.

Line 16 Service providers (doctors and nurses) experienced benefits as salary increasement, working with new equipment , opportunity for employment and education, etc. during the COVID-19 outbreak despite the fact that this period is characterized by more working hours and difficulties in patient management.

4) Line 41 "The Republic of Serbia was well", please provide de number in the ranking. 

Line 20 The Republic of Serbia was well ranked by the Global Health Security Index (59/195 and 33/43 for European region) which was supported by quick applying of healthcare measures during the first wave of COVID-19 pandemic [3,4].

5) Line 47 "and 3 new full-equipped " change it por "and three new full-equipped". 

Corrected , now line 26

6) Line 52 "It is concluded by many studies ...". Concluded by whom? if they conclude that, what is the novelty of this paper?

Studies from the reference list (3,5,6) analyzed literature on importance of establishing a relevant and sustainable motivation system for healthcare providers to optimize they work quality regardless of the pandemic conditions and especially during the pandemic. The novelty of this paper is that we analyzed healthcare quality from the aspect of healthcare workers via five quality dimensions and SERPERF methodology.

Results

7) Provide the central value and dispersion value of the age of both groups, also compare it with a T-student o Mann-Whitney U test (depending on their distribution). Is more powerful to compare the age as a continuous variable than a categorical. Also change the Figure 1 for continuous data.

Provided, Table 1. 

8) Same modifications for the income that are asked in the point #7 (for text and figure), es more accurate the analysis of the continuous data than the income as a categorical data. 

Provided, Table 1

9) Make a more clear, deep and critical interpretation of the results from the line 143-158.

Corrected, lines from 133-143

10) Improve de quality of the figure 3, the legends are not clear. 

Provided

Discussion

11) Make a more comparative analysis also provide a deeper explanation of the consequences of the results. The discussion is very superficial.

Provided, line 196-198,  201-213, 222-227, 228-231, 236,242, 261-265.

New references are added 30-32.

Round 2

Reviewer 2 Report

Some typos needs to correct. 

Line 220 should read 'this is a unique study'

Line 247 should read 'study is a consequence of pandemic,'

Line 261 the word 'is' is not needed.

Author Response

Dear reviewer,

Thank You for Your comments and improving the manuscript.

Reply to Your Comments:

Line 220 should read 'this is a unique study'- corrected

Line 247 should read 'study is a consequence of pandemic,'-corrected

Line 261 the word 'is' is not needed.-corrected

Kind regards,

Olivera Ivanov

Reviewer 3 Report

Rewrite the table 1 to the format of the journal. 

Author Response

Dear reviewer,

The table 1 is corrected to the format of the journal.

Thank You for Your comment and improving the quality of the manuscript.

Kind regards,

Olivera Ivanov